# NdFeB Magnets Recycling Process: An Alternative Method to Produce Mixed Rare Earth Oxide from Scrap NdFeB Magnets

Elif Emil Kaya [1,2,3], Ozan Kaya [4], Srecko Stopic [1,*], Sebahattin Gürmen [2] and Bernd Friedrich [1]

1    IME Process Metallurgy and Metal Recycling, RWTH Aachen University, 52056 Aachen, Germany; emil@tau.edu.tr (E.E.K.); bfriedrich@ime-aachen.de (B.F.)

2    Department of Metallurgical & Materials Engineering, Istanbul Technical University, 34469 Istanbul, Turkey; gurmen@itu.edu.tr

3    Department of Materials Science and Technology, Turkish-German University, 34820 Istanbul, Turkey

4    Department of Mechatronics Engineering, Istanbul Technical University, 34469 Istanbul, Turkey; kayaozan@itu.edu.tr

\*    Correspondence: sstopic@ime-aachen.de; Tel.: +49-176-7826-1674

**Abstract:** Neodymium iron boron magnets (NdFeB) play a critical role in various technological applications due to their outstanding magnetic properties, such as high maximum energy product, high remanence and high coercivity. Production of NdFeB is expected to rise significantly in the coming years, for this reason, demand for the rare earth elements (REE) will not only remain high but it also will increase even more. The recovery of rare earth elements has become essential to satisfy this demand in recent years. In the present study rare earth elements recovery from NdFeB magnets as new promising process flowsheet is proposed as follows; (1) acid baking process is performed to decompose the NdFeB magnet to increase in the extraction efficiency for Nd, Pr, and Dy. (2) Iron was removed from the leach liquor during hydrolysis. (3) The production of REE-oxide from leach liquor using ultrasonic spray pyrolysis method. Recovery of mixed REE-oxide from NdFeB magnets via ultrasonic spray pyrolysis method between 700 °C and 1000 °C is a new innovative step in comparison to traditional combination of precipitation with sodium carbonate and thermal decomposition of rare earth carbonate at 850 °C. The synthesized mixed REE- oxide powders were characterized by X-ray diffraction analysis (XRD). Morphological properties and phase content of mixed REE- oxide were revealed by scanning electron microscopy (SEM) and Energy-dispersive X-ray (EDX) analysis. To obtain the size and particle size distribution of REE-oxide, a search algorithm based on an image-processing technique was executed in MATLAB. The obtained particles are spherical with sizes between 362 and 540 nm. The experimental values of the particle sizes of REE-oxide were compared with theoretically predicted ones.

**Keywords:** rare earth elements; recycling; NdFeB; magnet; ultrasonic spray pyrolysis

## 1. Introduction

Rare earth elements (REEs) have a wide range of uses in technological products and applications. Due to the increased demand and supply risk, most REEs have been added to the list of critical metals. The production of REEs from primary resource causes environmental problems [1]. The recovery of REEs from waste materials is the most suitable strategy to find the solution of environmental problems and ensure the sustainability for production of REE raw materials in the future, according to an increased demand in industrial application. Most developed countries are importing REEs from China; 95% of REEs are supplied from China and in addition to this situation, export quotas of REEs applied by China have increased the export prices of REEs [2].

In order to produce rare earth elements oxides (REE-oxides), most researchers have studied different hydrometallurgical and pyrometallurgical strategies such as dry digestion [3], acid baking processes [4] and carbothermal reduction of ores and concentrates

with subsequent leaching using strong acids [5] aiming at higher REE extractions. Demol et al. [4] found that sulfation reaction of monazite with acid was virtually complete after baking at 250 °C for 2 h, resulting in >90% solubilization of REEs, thorium and phosphate. To prevent silica gel formation and to increase the extraction efficiency of REEs, before leaching, the dry digestion process was performed with concentrated HCl [6]. In contrast to application of an acid baking, Ma et al. reported [7] that rare earth recovery from eudialyte concentrate is achieved by avoiding silica-gel formation using a dry digestion process at room temperature. Generally, a direct leaching process was also applied for the treatment of red mud to obtain a high REE extraction efficiency [8,9]. Because of the many disadvantages of direct leaching processes such as high consumption of leaching agents and non-selectivity [10], Borra et al. [11] reported that alkali roasting–smelting–leaching processes allow the recovery of aluminum, iron, titanium, and REEs from bauxite residue. Generally, recovery of REEs from secondary materials is a new possibility for production of these critical metals.

Therefore, recycling has considerable advantages over processing natural ores and concentrates on account of energy effectiveness and selectivity [12]. Neodymium iron boron magnets (NdFeB) are the most valuable REE secondary resource because they contain a high content (approximately 20%) of REEs, neodymium (Nd), dysprosium (Dy) and some REEs in minor quantities, such as praseodymium (Pr). Between 20 and 25% of REEs produced worldwide are used in the production of NdFeB. Increasing future production of hard disks, automotive applications, motors, speakers, air conditioners, electronic devices, electric bicycles and wind turbines provides a strong driving force for finding a new process for recycling spent NdFeB magnets [13–15]. Furthermore, an alternative product that can replace NdFeB magnets in today's technologies in terms of performance and cost has not been developed yet. Therefore, the recycling of spent NdFeB magnets is the most promising effective alternative for the solution of the supply problem of Nd, Dy and Pr.

Önal et al. [16] studied recycling of NdFeB magnets using sulfation, selective roasting and water leaching, enabling the production of a liquid with at least 98% rare earth purity. Furthermore, 98% extraction efficiency of REEs from NdFeB magnets was obtained by the acid-baking process with nitric acid [17]. After the acid baking process and subsequent water leaching of the treated concentrate, the produced suspension was filtrated in order to separate a pregnant leaching solution. To produce the REE oxides from leach liquor, all the proposed methods in the literature are completely based on precipitation methods by using various precipitation agents such as sodium carbonate and oxalic acid [18,19].

It is known that REE-carbonate or REE-oxalate can be produced from impurities present in sulfuric liquors using oxalic acid and sodium carbonate by a precipitation method [20,21]. It was reported that high purity REE-oxide (99.2%) was achieved using oxalic acid as a precipitation agent. Relatively lower purity RE-oxide was produced using sodium carbonate during precipitation [18]. The precipitation behavior of REEs with precipitation agents including oxalate, sulfate, fluoride, phosphate, and carbonate was examined using thermodynamic principles and calculations [22]. It was found that the pH of the system, types of the precipitation agent and present anions in the leach liquor have a noteworthy impact on the purity of the REE precipitants.

In contrast to the precipitation method, the production of nanosized REEs using an ultrasonic spray pyrolysis method is missing in the literature. Ultrasonic spray pyrolysis (USP) combines the ultrasound used for dispersing the precursor solution into droplets and chemical decomposition of the dissolved material inside the droplets at elevated temperatures, resulting in the formation of fine metallic, oxidic and composite powder [23–25]. This technique has been successfully used in the production of REE-oxide, the results of which are $Y_2O_3$, $La_2O_3$ $Gd_2O_3$, and $CeO_2$ [26–29]. The USP method enables synthesized spherical and fine REE-oxide in one-step. Moreover, the technique is capable of metal oxide with controllable chemical composition, particle size and morphology of particles by manipulating process parameters, which for the precursor type and concentration, reaction atmosphere, carrier gas flow rate, and reaction temperature [30–32]. In the present

study, a new sustainable method was proposed for the production of mixed REE-oxide from REE-rich leach liquor. This proposed work summarizes the following operations: 1. Grinding and sieving; 2. Acid baking; 3. Calcination; 4. Leaching with water and 5. Ultrasonic spray pyrolysis, as shown at Figure 1.

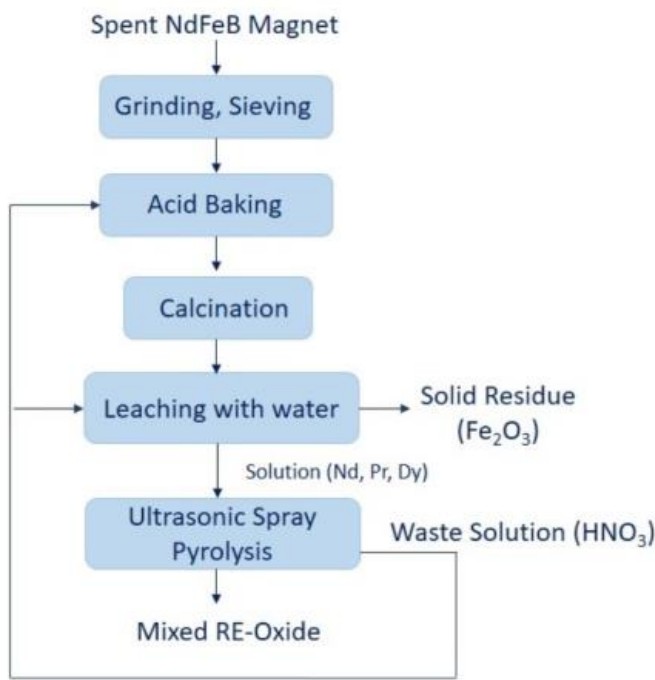

**Figure 1.** The proposed strategy for preparation of REE- oxides from spent NdFeB magnets.

In the final step of ultrasonic spray pyrolysis, the produced nitric acid shall be recycled and sent to the acid baking process. This study aims at investigating the conditions required to produce mixed REE-oxides in the combined hydrometallurgical process (acid baking with water dissolution and ultrasonic spray pyrolysis process). A literature review reveals that this information is currently not reported for the production of mixed RE-oxide using leach liquor. The proposed route promotes the enhancement of the circular economy of critical raw materials/REEs and could provide a high potential to increase resource efficiency for spent NdFeB magnets.

## 2. Experimental

### 2.1. Materials, Acid Baking, and Water Leaching

Waste NdFeB magnets used during the experiments were supplied in bulk form. Demagnetization was not necessary. Bulk and brittle NdFeB magnet pieces were crushed by jaw crusher Retsch BB 50, (Retsch GmbH, Haan, Germany) using dry ice to prevent magnet powders from catching fire. The crushing process was repeated three times to obtain the magnet powders to suitable powder's size. Nitric acid (65%) was used for acid baking without dilution and was purchased from VWR International GmbH, Darmstadt, Germany in analytical grade. All reagents were used without further purification. All solutions were prepared using deionized water. 16.6-gram magnet powders were dissolved in 500 mL of 2 molar $HNO_3$ acid solution to determine the chemical composition of the magnets. The chemical analysis of obtained solution was performed using ICP-OES analysis (SPECTRO ARCOS, SPECTRO Analytical Instruments GmbH, Kleve, Germany). Elemental composition of the NdFeB was determined by X-ray fluorescence (XRF) spectroscopy (Panalytical WDXRF spectrometer (Malvern Panalytical B.V., Eindhoven, The Netherlands)).

The extraction of REEs from NdFeB magnets was performed by nitric acid baking and a subsequent water leaching. The acid baking process was employed using highly

concentrated $HNO_3$ (65%) with a 1:5 solid/liquid (S/L) ratio. Water was first added to the magnet powders to promote the ionization of the nitric acid before the acid baking process. After waiting 1 h, the mixture was calcined at 200 °C for 2 h. Water leaching experiments were performed in a 500 mL four-neck glass reactor equipped with a heating mantle and temperature controller (IKA Werke GmbH, Staufen im Breisgau, Germany) The leaching solution was kept under 550 revolutions/min agitation by a mechanical stirrer. Water leaching experiments were conducted with a 1:15 solid/liquid (S/L) ratio for 90 min. The leaching mixture was filtered using the filtering set up to separate the leaching solution from leach residue. Chemical content in the leach liquor was analyzed by ICP-OES to determine the purity of the leaching solution containing REE. The theoretical background of acid baking with nitric acid and water leaching process was reported elsewhere [17].

## 2.2. Ultrasonic Spray Pyrolysis Method for Production of RE-Oxide and Their Characterization

Very fine aerosol droplets were obtained from a leach solution using an ultrasonic atomizer (PRIZNano, Kragujevac, Serbia), with a frequency 1.75 MHz in an ultrasonic field obtained by 3 ultrasonic transducers. The aerosol was carried with nitrogen flow rate 1.0 L/min into in quartz tube (1.0 m length and 0.021 m diameter) between 700 °C and 1000 °C, placed in a Ströhlein Furnace, Selm, Germany. The flow rate of nitrogen was measured using special flowmeter gas unit (YOKOGAWA Deutschland GmbH, Ratingen). One step ultrasonic spray pyrolysis lab-scale horizontal equipment was shown in Figure 2. Experimental parameters were given in Table 1.

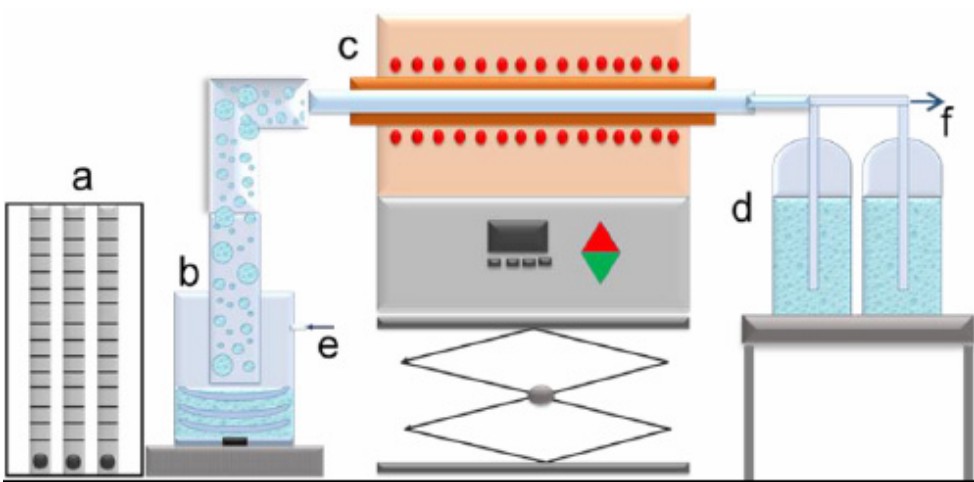

**Figure 2.** One step ultrasonic spray pyrolysis lab-scale horizontal equipment: (**a**)—gas flow regulation; (**b**)—ultrasonic aerosol generator; (**c**)—furnace with the wall heated reactor; (**d**)—collection bottles; e—gas inlet, f—gas outlet.

**Table 1.** Experimental parameters of ultrasonic spray pyrolysis method.

| Samples Codes | Concentration of $Nd(NO_3)_3$ (g/L) | Concentration of $Pr(NO_3)_3$ (g/L) | Concentration of $Dy(NO_3)_3$ (g/L) | Reaction Temp (°C) | $N_2$ Flow Rate (L/min) | Ultrasonic Frequency (MHz) |
|---|---|---|---|---|---|---|
| S1 | 0.458 | 0.130 | 0.010 | 700 | 1.0 | 1.75 |
| S2 | 0.458 | 0.130 | 0.010 | 800 | 1.0 | 1.75 |
| S3 | 0.458 | 0.130 | 0.010 | 900 | 1.0 | 1.75 |
| S4 | 0.458 | 0.130 | 0.010 | 1000 | 1.0 | 1.75 |

The SEM analysis of particles obtained by ultrasonic spray pyrolysis was performed at JSM 7000F by JEOL, (Construction year 2006, Japan) and EDX-analysis using Octane Plus-A by Ametek-EDAX, (construction year, 2015, USA) with Software Genesis V 6.53 by Ametek. XRD Analysis of RE-oxides powders was performed using Bruker D8 Advance

with LynxEye detector (Bruker AXS, Karlsruhe, Germany). X-ray powder diffraction patterns were collected on a Bruker-AXS D4 Endeavor diffractometer in Bragg–Brentano geometry, equipped with a copper tube and a primary nickel filter providing Cu $K\alpha_{1,2}$ radiation ($\lambda$ = 1.54187 Å).

## 3. Results and Discussion

Mixed RE-Oxide powders were synthesized by a one-step USP method from leach liquor. Thermodynamic investigations of a possible reaction were conducted by HSC software package 6.12 (Outotec, Espoo, Finland). Various reaction temperatures from 700 °C to 1000 °C were tested to investigate their role on the phase formation of RE-Oxide. The mixed REE-Oxide powders were characterized by X-ray diffraction analysis, scanning electron microscopy. To reveal size and size distribution of mixed RE-Oxide, SEM micrographs were examined via image-processing techniques in MATLAB (MathWorks, Natick, MA, USA).

### 3.1. Characterization of Scrap NdFeB Magnet

The magnet composition was determined by X-ray fluorescence (XRF). The major elements of the NdFeB magnet powder are Fe, Nd, Pr as major elements and the trace amounts of Mn, Co, Pd, Al and Si, as shown in Table 2.

**Table 2.** Chemical composition of NdFeB magnet powders determined by XRF.

| Composition | $Na_2O$ | $Al_2O_3$ | $SiO_2$ | MnO | $Fe_2O_3$ | $Co_3O_4$ | CuO |
|---|---|---|---|---|---|---|---|
| Concentration (%) | 0.34 | 0.42 | 0.24 | 1.97 | 68.1 | 0.70 | 0.14 |
| **Composition** | $Ga_2O_3$ | $As_2O_3$ | $Nb_2O_5$ | PdO | $Pr_2O_3$ | $Nd_2O_3$ | $Tb_4O_7$ |
| Concentration (%) | 0.20 | 0.21 | 0.12 | 0.24 | 5.72 | 20.4 | 0.70 |

The contents of the NdFeB magnets were measured using inductively coupled plasma optical emission spectroscopy (ICP-OES). The ICP-OES analysis results of NdFeB magnet is given in Table 3.

**Table 3.** Chemical composition of NdFeB magnet powders sample.

| Composition | B | Co | Cr | Cu | Dy |
|---|---|---|---|---|---|
| Concentration (mg/L) | 278 | 245 | <1 | 32.6 | 210 |
| **Composition** | Fe | Mo | Nd | Ni | Pr |
| Concentration (mg/L) | 210,000 | <1 | 7580 | <1 | 2340 |

ICP-OES analysis showed the presence of Fe, Nd, and Pr as the major elements and Cu and Co as minor elements.

X-ray diffraction (XRD) analyses were conducted to identify the phases in the NdFeB magnet powders. XRD analysis results of NdFeB magnet powders were given in Figure 3.

According to XRD patterns, the powder sample was well crystallized in the $Nd_2Fe_{14}B$ phase. The X-ray diffraction peaks could be indexed to the tetragonal structure with space group P42/mnm (JCPDS card 40-1028).

Scanning Electron Microscopy (SEM) analyses were performed to observe the morphology of the NdFeB magnet powders, as shown at Figure 4.

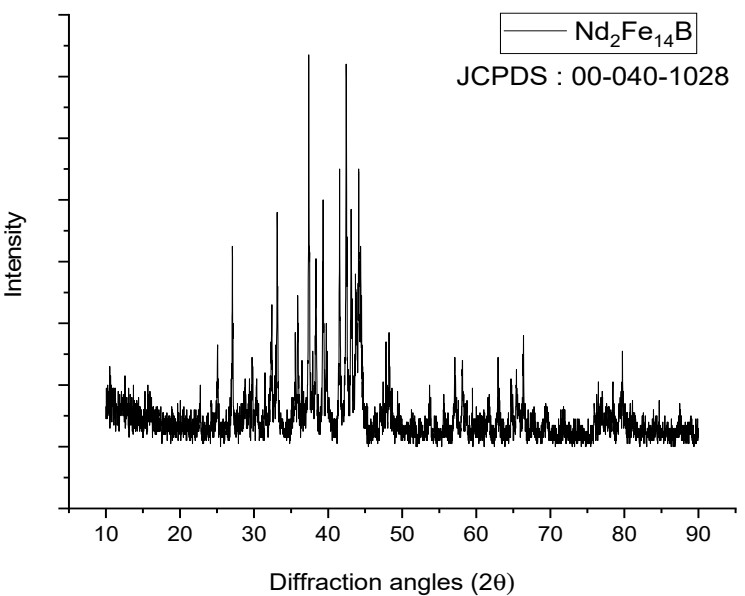

**Figure 3.** XRD pattern of NdFeB magnet powders.

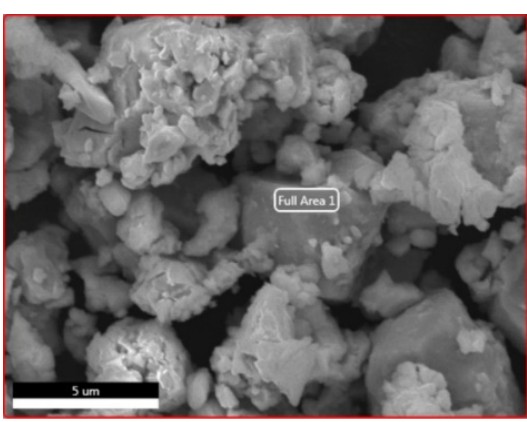

**Figure 4.** SEM analysis of NdFeB magnet powders.

As shown at Figure 5, Energy Dispersive Spectroscopy (EDS) results demonstrate that NdFeB magnet powders primarily consist of Fe and Nd. EDS results are in good agreement with the ICP analysis results but due to the small amount of the other elements, they cannot be detected by EDS.

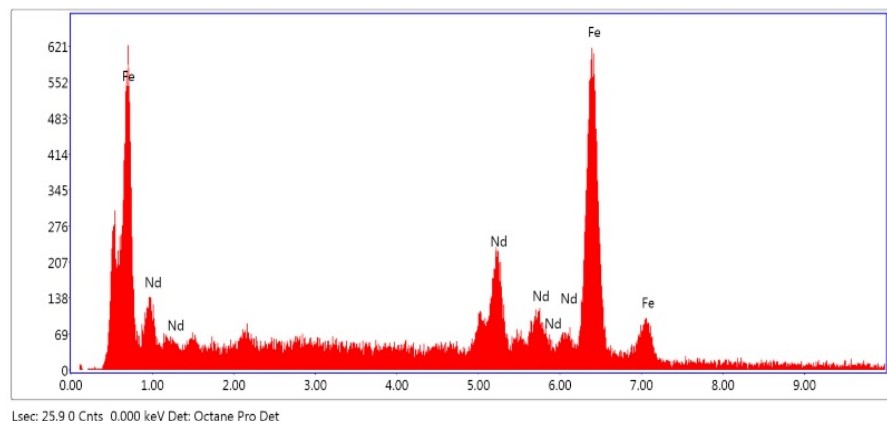

**Figure 5.** EDS-analysis of NdFeB magnet powders.

### 3.2. Production of REE-Oxide and Their Characterization

The production of mixed REE-oxides powder from scrap NdFeB magnet by nitric acid baking and water leaching followed by ultrasonic spray pyrolysis method was investigated. The concentration of metals ions in the leached solution were determined using ICP-OES analysis. The chemical composition of leach liquor obtained after the water leaching process is illustrated in Table 4. Leach liquor of the same chemical composition was used in all USP experiments.

**Table 4.** Chemical composition of the leach liquor.

| Composition | B | Co | Cr | Cu | Dy |
|---|---|---|---|---|---|
| Concentration (mg/L) | 80 | 30 | <1 | <1 | 100 |
| **Composition** | **Fe** | **Mo** | **Nd** | **Ni** | **Pr** |
| Concentration (mg/L) | <1 | < 1 | 4580 | <1 | 1300 |

Gibbs free energy change depending on reaction temperature was computed by HSC software (Outotec, Espoo, Finland), as shown at Figure 6. The formation of RE-oxides after evaporation of water in the furnace can be described as in the following equations:

$$2Nd(NO_3)_3 = Nd_2O_3 + 6NO_2 + 1.5O_2 \tag{1}$$

$$2Dy(NO_3)_3 = Dy_2O_3 + 6NO_2 + 1.5O_2 \tag{2}$$

$$2Pr(NO_3)_3 = Pr_2O_3 + 6NO_2 + 1.5O_2 \tag{3}$$

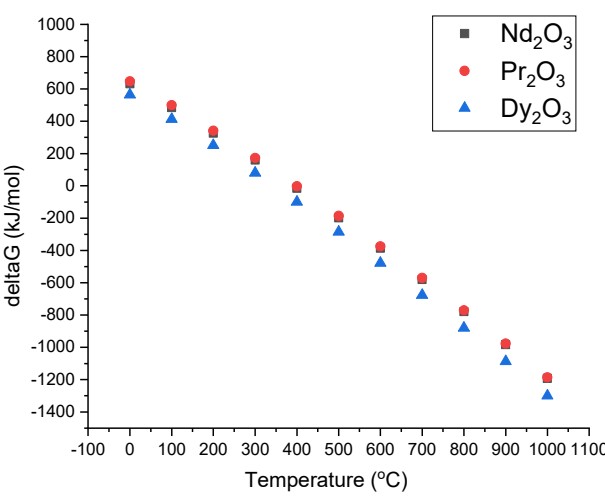

**Figure 6.** Gibbs free energy change depending on reaction temperature.

The Gibbs free energy for the temperature range of 0–1000 °C is exhibited in Figure 6. As can be seen, the Gibbs free energy is negative after 500 °C. This allows for RE-oxide to be formed by the thermal decomposition of leach liquor, which is energetically favored after 500 °C.

Figure 7 shows the XRD results for the samples synthesized at 700 °C, 800 °C, 900 °C and 1000 °C by ultrasonic spray pyrolysis method.

XRD analysis of powders obtained between 700 °C and 1000 °C confirmed the formation of a mixture of RE-oxides. The cubic structure of $Nd_2O_3$ with 20% of $Pr_2O_3$ was found between 700 °C and 800 °C as shown at Figure 7. Checks of the XRD Pattern for crystal structure leads to $Nd_{1.6}Pr_{0.4}O_3$ solid solution. An increase in temperature from 800 °C to 900 °C and 1000 °C leads to a mixture of cubic and trigonal structure of $Nd_2O_3$ with 20% of $Pr_2O_3$, as shown at Figure 7. An increase in temperature from 700 °C to 1000 °C increases

the crystallinity of the obtained structure. Additionally, typical EDX-Analysis of powders was shown at Figure 8, confirming the presence of rare earth elements.

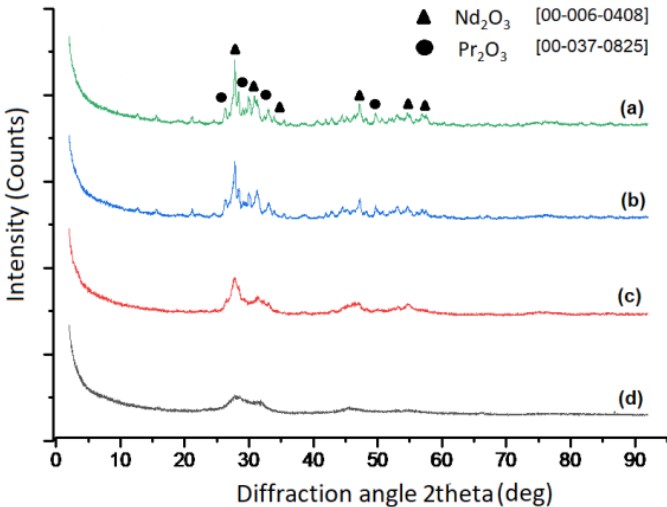

**Figure 7.** XRD analysis of RE-oxide powders synthesized with varying reaction temperatures (**a**) 1000 °C, (**b**) 900 °C, (**c**) 800 °C and (**d**) 700 °C.

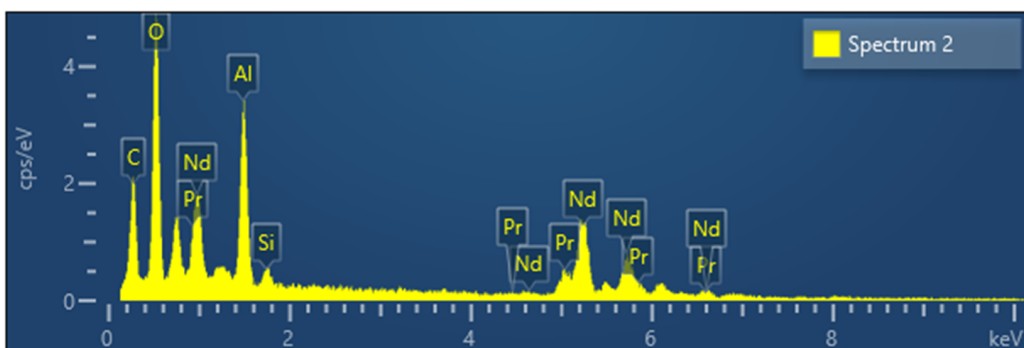

**Figure 8.** Typical EDS analyses of mixed REE-Oxide powders synthesized with varying reaction temperatures.

The morphological investigation of mixed RE-oxide produced by USP processes at different temperatures was conducted by SEM analysis. SEM analyses of the RE-oxide are illustrated in Figure 9.

As indicated in Figure 9, spherical RE-oxide was obtained at various reaction temperatures by the USP Method. The image-processing technique is one of the computational approaches widely getting implemented in various fields of material science. It is especially useful for interpreting the images as the results of SEM. The morphology and size of the RE-oxide nanoparticles were analyzed by SEM. Using SEM results, the morphological features of the RE-oxide nanoparticles, such as their diameter, were picked up by image processing and a particle search algorithm. The use of an image processing method algorithm is detailed in [32].

Applied image processing methods generate the black and white images from the original SEM images for determining the location and size of the RE-oxide nanoparticles. Since the particles are known to have spherical shapes [33], the Hough transform method was utilized for approximately defining the nanoparticles. The Hough transform draws new circles at the three boundary points. Then, the center of the circle is computed, with the junction point of new circles and diameter limits defined by the user. After the detecting RE-oxide nanoparticles, these particles were labelled with blue rough circles and their

cumulative distribution results related to process conditions were achieved. The results are given in Figure 10.

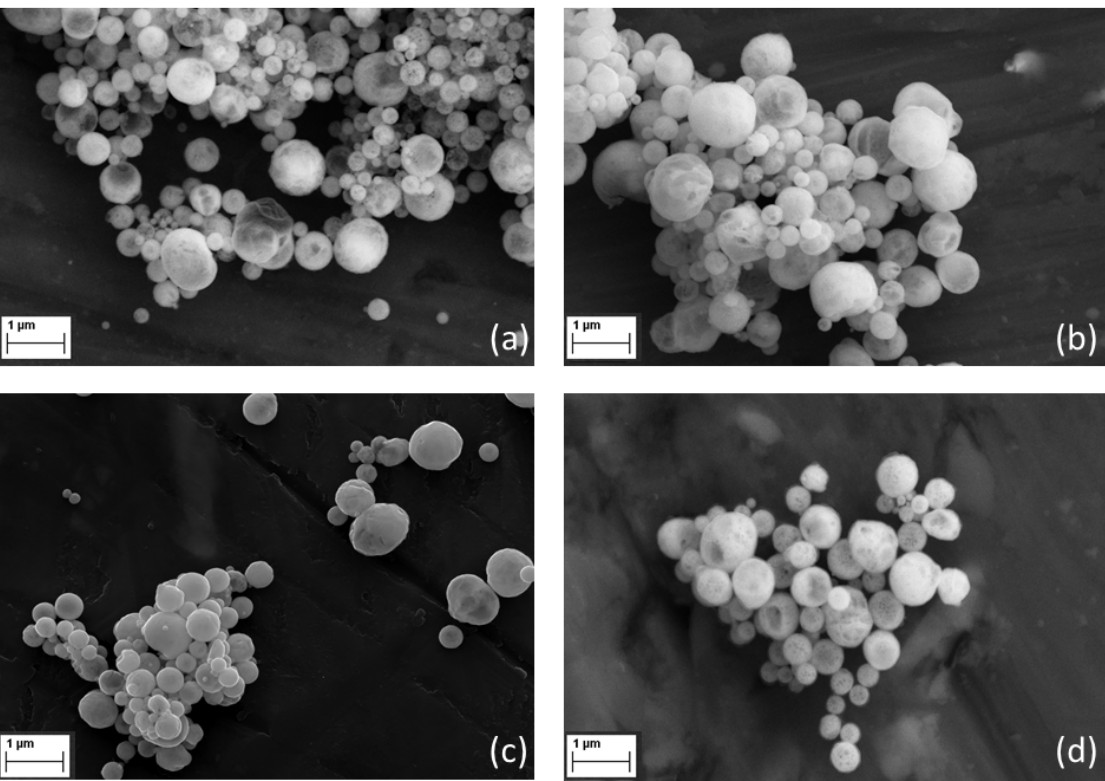

**Figure 9.** SEM analyses of mixed RE-Oxide powders synthesized with varying reaction temperatures (**a**) 700 °C; (**b**) 800 °C; (**c**) 900 °C; (**d**) 1000 °C.

Graph of the labelled nanoparticles were drawn, with cumulative distribution represented by the *y*-axis, and nanoparticle size represented by the *x*-axis, as seen in Figure 10a–d. The cumulative curve of RE-oxide nanoparticles whose sizes were calculated by an image-processing technique is represented by the blue dashed line. The mean values of RE-oxide nanoparticle size were calculated from SEM by the image-processing technique. These SEM results reveal that the particles of RE-oxide synthesized from a 0.6 g/L solution concentration at various reaction temperatures lay in the range of 200–700 nm. The mean particle size of RE-oxide synthesized at 700 °C, 800 °C, 900 °C and 1000 °C was found to be 362 nm, 417 nm, 468 nm and 540 nm, respectively.

The theoretical particle size of RE-oxides was calculated according to related equations. The formation of RE-oxides will be firstly defined via the diameter of aerosol droplet ($d_d$) as shown with Equation (4) proposed by Peskin and Raco [24]:

$$d_d = 0.34 \left( \frac{8\,\pi\sigma}{\rho_L f^2} \right)^{\frac{1}{3}} \tag{4}$$

where: $f$—ultrasound frequency; $\rho_L$—density of water solution; $\sigma$—surface tension.

Using the following values: for water solution: $f$- 1.75 MHz; $\rho_L$- 1.02 g/cm$^3$; $\sigma$- 0.07 J/m$^2$, the calculated aerosol droplet amounts 2.86 μm.

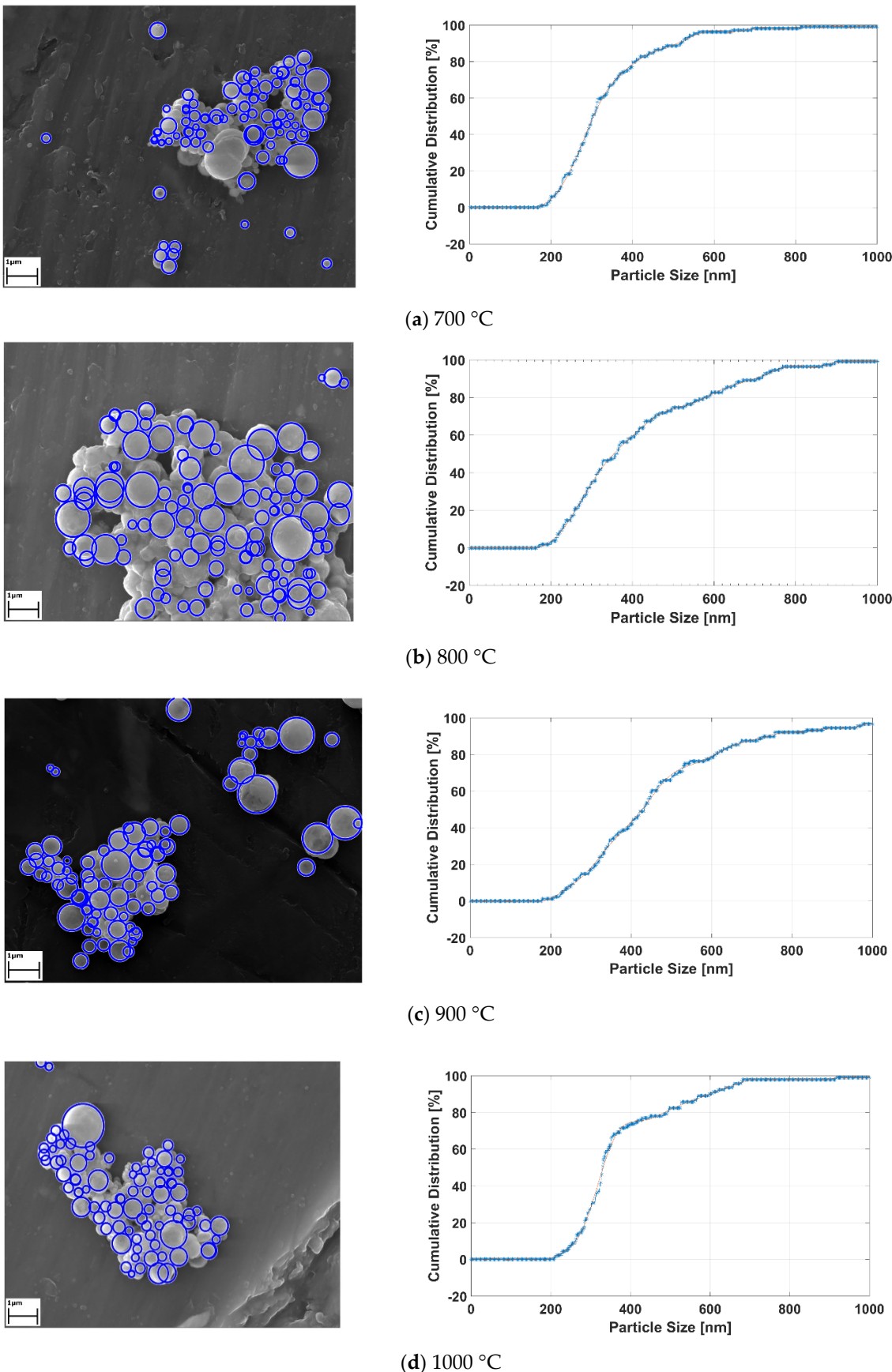

**Figure 10.** The detecting RE-oxide nanoparticles and theirs' cumulative curves related to USP-experiment conditions at (**a**) 700 °C, (**b**) 800°C, (**c**) 900°C and (**d**) 1000 °C.

The particle size *(dp)* depends on the droplet size and the concentration of the solution (*C*). This correlation between the concentration and other precursor characteristics and the final particle size, under the assumption that no precursor is lost in the process, can be described with the following Equation (5) derived via the Equation by Messing et al. [25]:

$$d_p = d_d \left( \frac{M_{REE-Nitrate}}{M_{REE-Oxide}} * \frac{C}{\rho} \right)^{0,33} \tag{5}$$

where the *dp* is the diameter of the particle, the $d_d$ is the diameter of the aerosol droplet calculated with Equation (4), Mp-molar mass of REE-nitrate (g/mol), the $\rho$ is the density of REE-Oxide (Nd-, Pr-, and Dy-oxide), *C* is the concentration of the precursor solution.

Using the following values for molar mass of rare earth elements – nitrate (REE-nitrate) and rare earth elements- oxides (REE-oxides), densities of REE-oxides and concentrations of metals in solution (as shown in Table 4), the calculated values for particles sizes of REE-oxides using Equation (5) are presented in Table 5.

**Table 5.** Calculated theoretical particle size of RE-oxides using Equation (5).

| **REE-Nitrate** | **Nd(NO$_3$)$_3$** | **Pr(NO$_3$)$_3$** | **Dy(NO$_3$)$_3$** |
|---|---|---|---|
| Molar mass of REE-nitrate (g/mol) | 282.2 | 326.0 | 348.5 |
| REE-oxides | Nd$_2$O$_3$ | Pr$_2$O$_3$ | Dy$_2$O$_3$ |
| Density (g/cm$^3$) | 7.2 | 6.9 | 7.8 |
| Molar mass (g/mol) | 336.5 | 329.8 | 373.0 |
| Concentration of metal in solution (g/L) | 0.458 | 0.130 | 0.010 |
| Theoretical minimal particle size (nm) | 108 | 76 | 31 |

The calculated minimal particle size (nm) amounts: 108, 76, 31 for Nd$_2$O$_3$, Pr$_2$O$_3$ and Dy$_2$O$_3$, total 215. The obtained values of particle sizes are compared with experimentally obtained values obtained by image process techniques. The differences between calculated and experimentally obtained values may be partially due to the approximate values used for surface tension and density of water solution, and mostly due to coalescence/agglomeration of aerosol droplets during transport to the furnace from an aerosol generator. Moreover, Equation (5) was based on the assumption of one particle per one droplet, and the influence of temperature on the mean particle size between 700 °C and 1000 °C was not taken into consideration.

## 4. Conclusions

Spherical particles of REE-oxides were produced from spent NdFeB magnets using a combined process and consists of: nitric acid baking process at 200 °C, water leaching, and ultrasonic spray pyrolysis between 700 °C and 1000 °C. Iron was removed from water solution using a hydrolysis process. XRD analysis of the obtained particles found a cubic and trigonal structure Nd$_2$O$_3$ with 20% Pr$_2$O$_3$, which is according to detected stoichiometry in solution after dissolution of spent NdFeB magnets. An increase in temperature from 700 °C to 1000 °C increases not only the crystallinity of the structure, but also the particle size between 362 and 540 nm. The minimal theoretical total particle size of prepared REE-oxides amounts to 215 nm. The differences between calculated and experimentally obtained values may be partially due to coalescence/agglomeration of aerosol droplets during transport to the furnace from an aerosol generator. Generally, we developed one combined environmentally friendly process for recovery of nanosized powder mixture of Nd$_2$O$_3$ and Pr$_2$O$_3$ from spent magnets and re-use of nitric acid. The final winning of the mixture of metallic Nd and Pr will be ensured using molten salt electrolysis [34].

**Author Contributions:** Conceptualization, E.E.K. and S.S.; funding acquisition, B.F. and S.G.; investigation, E.E.K.; methodology, E.E.K. and S.S.; supervision, S.G. and B.F.; writing—original draft, E.E.K., O.K. and S.S. All authors have read and agreed to the published version of the manuscript.

**Funding:** The research leading to these results has received funding from the AIF- German Federation of Industrial Research Associations, Germany and TÜBITAK- The Scientific of Technological Research Council of Turkey (Call identifier CORNET 29th Call) under grant agreement EN03193/20). The authors would like to greatly acknowledge TUBITAK/Turkey (Project No: 120N331) for financial support. Elif Emil Kaya would like to thank DAAD "Research stays of doctoral research assistants of the TDU at German partner universities" for financial support.

**Institutional Review Board Statement:** Not applicable.

**Informed Consent Statement:** Not applicable.

**Data Availability Statement:** Not applicable.

**Conflicts of Interest:** The authors declare no conflict of interest.

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
