# Peer review of "NdFeB Magnets Recycling Process: An Alternative Method to Produce Mixed Rare Earth Oxide from Scrap NdFeB Magnets"

_metals, doi:10.3390/met11050716_

Round 1

Reviewer 1 Report

The manuscript requires substantial grammar editing.

In the experimental methods, the authors need to state the concentrations of their dilute and concentrated nitric acid solutions.

In Figure 5, there are two curves both marked "c". Presumably one should be "d"?

On page 12, the authors provide two equations, equation 4 and 5. The values calculated with eq.4 is used in eq. 5. However, in the text the authors referred to the values obtained with eq. 4 as "the values obtained with eq. 1. before they substitute it in eq. 5. This needs to be corrected.

Author Response

Response to Editor and Reviewers :

Firstly, the authors would like to thank to reviewer and editor for the collaboration in the improvement of the work. The manuscript was revised and corrections were made based on the reviewer’s and editor’s suggestions. The responses to the comments of the reviewers and editor are listed next and the corrections made are highlighted in the revised manuscript.

1) The manuscript requires substantial grammar editing.

We thank the reviewer for pointing out this issue. The manuscript was edited for proper.

2) In the experimental methods, the authors need to state the concentrations of their dilute and concentrated nitric acid solutions.

Thanks to the editor for this comment. All explanations were added in the manuscript as below.

Nitric acid (65 %) was used for acid baking without dilution and was purchased from VWR International GmbH, Darmstadt, Germany in analytical grade.

16.6- gram magnet powders were dissolved in 500 ml of 2 molar HNO3 acid solution to determine the chemical composition of the magnets.

The acid baking process was employed highly concentrated HNO3 (65%) with a 1:5 solid/liquid (S/L) ratio

3) In Figure 5, there are two curves both marked "c". Presumably one should be "d"?

You have right. Thank you for your attention, it was corrected. We added new improved XRD-Analysis. 

4) On page 12, the authors provide two equations, equation 4 and 5. The values calculated with eq.4 is used in eq. 5. However, in the text the authors referred to the values obtained with eq. 4 as "the values obtained with eq. 1. before they substitute it in eq. 5. This needs to be corrected.

You have right. This is mistake. We changed it in our improved text.

Reviewer 2 Report

The manuscript described a process for the recovery of rare earth elements from scrap NdFeB magnets. The manuscript is poorly written with numerous grammatical errors. Most of the figures are poorly prepared. Formatting is poor. All of these demonstrates that the authors did not take the manuscript preparation seriously. The following lists grammatical errors that are found in the first three pages of the manuscript, in addition to further comments on the poor preparation of figures and tables. The reviewer does not want to point out every grammatical error in the whole manuscript, which is time-consuming and not enjoyable.

  1. Abstract, Lines 19-20, Grammatical error: “In the present study rare earth elements recovery from NdFeB magnets a promising process flowsheet is proposed as follows;”
  2. Abstract, Lines 21-22, the following sentence is confusing: “Applying of the water leaching to removal of the iron from the leach liquor during hydrolysis.”
  3. Abstract, Lines 27-28, Grammatical error: “Morphological properties and phase content of mixed rare earth oxide were revealed scanning electron microscopy 28 (SEM) and Energy-dispersive X-Ray (EDX) analysis.”
  4. Lines 39-41: “The recovery of REEs is the most suitable strategy to find the solution of environmental problems and ensure the sustainability of REE raw materials in future according to an increased demand in industrial application” The sentence is very confusing.
  5. Lines 59-60: “Generally, recovery of REEs from secondary materials is from high importance for ensuring of these critical metals”. Very confusing.
  6. Lines 71-73: “Therefore, the recycling of spent NdFeB magnets, whose service life is completed or which are scrapped due to a process error, is the most effective alternative for the solution of the supply problem of raw materials”. Very confusing.
  7. Lines 77-78: “After the acid baking process and subse-quently water leaching of the treated concentrate.” Incomplete sentence.
  8. Lines 84-85: “It was reported that high purity RE-oxide (99.2 %) was achieved using oxalic acid as a precipitation agent however, using sodium carbonate relatively lower purity RE-oxide was produced [18].” Grammatical error.
  9. Lines 91-92: “In contrast to the precipitation method, the production of nanosized REEs using ul-trasonic spray pyrolysis method is missing the literature” Grammatical error.
  10. Lines 102-103: “This methodology contains the next steps:” Grammatical error.
  11. Lines110-111: “The advantages of the proposed method are that it can be carried out in a short time and saves on external energy consumption” Weird expression.
  12. Lines 120-121: “Nitric acid (65 %) is used as a leaching agent were purchased from VWR International GmbH, Darmstadt, Germany in analytical grade.” Grammatical error.
  13. Lines 121-122: “All reagents used without further purification and all solutions were prepared with deionized water.” Grammatical error.
  14. Lines129-130: “The extraction of REEs from NdFeB magnets by acid baking with HNO3 followed by water leaching.” Grammatical error.
  15. Lines130-131:“The acid baking process was employed highly concentrated HNO3 with a 1:5 solid/liquid (S/L) ratio.” Grammatical error.
  16. Lines 131-132: “Some water added to the magnet powders to promote the ion-ization the nitric acid before acid-baking process.” Grammatical error.
  17. Lines 143-145: “Very fine aerosol droplets of previously hydrometallurgical prepared leach solution from waste magnets were obtained with an ultrasonic atomizer (PRIZNano, Kragujevac, Serbia),”Grammatical error.
  18. Image quality of Figure 1 is very poor. In addition, the authors should label each unit in the system with numbers and indicate what they are, rather than simply putting a picture with low resolution.
  19. In Figure 3, SEM and EDS images have different width, which looks really bad.
  20. Caption of Figure 4: “Thermodynamic analysis of the obtained leach liquor was.” Seriously?
  21. Figure 5, no labelling of any peak.
  22. Table 5 is put inside a box, which I have never seen before.

Author Response

Firstly, the authors would like to thank to reviewer and editor for the collaboration in the improvement of the work. The manuscript was revised and corrections were made based on the reviewer’s and editor’s suggestions. The responses to the comments of the reviewers and editor are listed next and the corrections made are highlighted in the revised manuscript

  1. Abstract, Lines 19-20, Grammatical error: “In the present study rare earth elements recovery from NdFeB magnets as new promising process flowsheet is proposed as follows;” “In the present study rare earth elements recovery from NdFeB magnets as new promising process flowsheet is proposed as follows;”
  2. Abstract, Lines 21-22, the following sentence is confusing: “Applying of the water leaching to removal of the iron from the leach liquor during hydrolysis.” Iron was removed from the leach liquor during hydrolysis.
  3. Abstract, Lines 27-28, Grammatical error: “Morphological properties and phase content of mixed rare earth oxide were revealed scanning electron microscopy 28 (SEM) and Energy-dispersive X-Ray (EDX) analysis.” “Morphological properties and phase content of mixed rare earth oxide were revealed by scanning electron microscopy 28 (SEM) and Energy-dispersive X-Ray (EDX) analysis.”
  4. Lines 39-41: “The recovery of REEs is the most suitable strategy to find the solution of environmental problems and ensure the sustainability of REE raw materials in future according to an increased demand in industrial application” The sentence is very confusing. "The recovery of REEs from waste materials is the most suitable strategy to find the solution of environmental problems and ensure the sustainability for production of REE raw materials in future according to an increased demand in industrial application” 
  5. Lines 59-60: “Generally, recovery of REEs from secondary materials is from high importance for ensuring of these critical metals”. Very confusing. Generally, recovery of REEs from secondary materials is new possibility  for production of these critical metals”. 
  6. Lines 71-73: “Therefore, the recycling of spent NdFeB magnets, whose service life is completed or which are scrapped due to a process error, is the most effective alternative for the solution of the supply problem of raw materials”. Very confusing.“Therefore, the recycling of spent NdFeB magnets,  is the most promising alternative for the solution of the supply problem of Nd, Dy and Pr”.
  7. Lines 77-78: “After the acid baking process and subsequently water leaching of the treated concentrate.” Incomplete sentence. After the acid baking process and subsequent water leaching of the treated concentrate, the produced suspension was filtrated in order to separate a pregnant leaching solution. 
  8. Lines 84-85: “It was reported that high purity RE-oxide (99.2 %) was achieved using oxalic acid as a precipitation agent however, using sodium carbonate relatively lower purity RE-oxide was produced [18].” Grammatical error. We changed it. "It was reported that high purity RE-oxide (99.2 %) was achieved using oxalic acid as a precipitation agent. Relatively lower purity RE-oxide was produced using sodium carbonate during precipitation.
  9. Lines 91-92: “In contrast to the precipitation method, the production of nanosized REEs using ul-trasonic spray pyrolysis method is missing the literature” Grammatical error. In contrast to the precipitation method, the production of nanosized REEs using ultrasonic spray pyrolysis method is missing in the literature.
  10. Lines 102-103: “This methodology contains the next steps:” Grammatical error. "This proposed work summarizes the following operations:"
  11. Lines110-111: “The advantages of the proposed method are that it can be carried out in a short time and saves on external energy consumption” Weird expression. We removed this sentence.
  12. Lines 120-121: “Nitric acid (65 %) is used as a leaching agent were purchased from VWR International GmbH, Darmstadt, Germany in analytical grade.” Grammatical error. Nitric acid (65 %) was used for acid baking without dilution and was purchased from VWR International GmbH, Darmstadt, Germany in analytical grade
  13. Lines 121-122: “All reagents used without further purification and all solutions were prepared usin deionized water.” Grammatical error. All reagents were used without further purification. All solutions were prepared using deionized water
  14. Lines129-130: “The extraction of REEs from NdFeB magnets by acid baking with HNO3 followed by water leaching.” Grammatical error.“The extraction of REEs from NdFeB magnets was performed by nitric acid baking  and a subsequent water leaching
  15. Lines130-131:“The acid baking process was employed highly concentrated HNO3 with a 1:5 solid/liquid (S/L) ratio.” Grammatical error. The acid baking process was employed using highly concentrated HNO3 with a 1:5 solid/liquid (S/L) ratio.”  
  16. Lines 131-132: “Some water added to the magnet powders to promote the ion-ization the nitric acid before acid-baking process.” Grammatical error. Water was firstly added to the magnet powders to promote the ionization of the nitric acid before acid-baking process
  17. Lines 143-145: “Very fine aerosol droplets of previously hydrometallurgical prepared leach solution from waste magnets were obtained with an ultrasonic atomizer (PRIZNano, Kragujevac, Serbia),”Grammatical error. Very fine aerosol droplets were obtained from a leach solution using an ultrasonic atomizer (PRIZNano, Kragujevac, Serbia)
  18. Image quality of Figure 1 is very poor. In addition, the authors should label each unit in the system with numbers and indicate what they are, rather than simply putting a picture with low Resolution. We put new picture with high Resolution and explained our experimental Setup with a, b, c, d, e, f. Figure 1: One step ultrasonic spray pyrolysis lab-scale horizontal equipment: a- gas flow regulation; b- ultrasonic aerosol generator; c- furnace with the wall heated reactor; d- collection bottles; e- gas inlet, f- gas outlet
  19. In Figure 3, SEM and EDS images have different width, which looks really bad. You have right. We changed with for both pictures. Now it looks better.
  20. Caption of Figure 4: “Thermodynamic analysis of the obtained leach liquor was.” Seriously?  As suggested, it was corrected as follows. Gibbs free energy change depending on reaction temperature was computed by HSC software (Outotec, Espoo, Finland), as shown at Fig. 4. 

    Figure 4. Gibbs free energy change with temperature for RE-oxide

  21. Figure 5, no labelling of any Peak. We put new Picture with labelling of Peaks.
  22. Table 5 is put inside a box, which I have never seen before. We changed it.

Reviewer 3 Report

the great problem with your paper is that your process does not give a separation of the elements in the magnet waste. You process will always give a final product which is the average of the elemental composition of all the magnets which went into the process. I think as a result it is not a good process, if a few magnets which are samarium cobalt were to be accidently added to the neodynium / iron magnets which the process is fed on then the resulting magnet powder from your process would be contaminated with cobalt and samarium. I suspect that the addition of cobalt / samarium to the magnets would have an effect on their magnetic properties.

Without a means of making a separation the recycling industry will make new products which will be lower and lower grade. I know that the Prince of Wales is concerned that nanobots will reduce the earth to "grey goo" but while Charles is concerned about a problem which does not exist. I think that your process will make in the long run the magnet version of "grey goo", each generation of magnets from the recycling process will be worse than the last.

I think that what you need is a separation method which allows you to improve the purity of the Fe and Nd in the solid which is going to be made into new magnets.

figure 2 has a lot of noise, the quaility of the data and the figure are not very convincing to me. I think that you need to work on the quaility of the XRD data which you present.

your paper is not easy to follow, in figure 5 you show XRD traces for solids made at different tempertures. Rather than writing the tempertures on the graph you use "cryptic codes". Please put an effort in to make the paper more easy to read. You the author should put the effort in to making the paper easy to read rather than expecting the reading to make an effort to deal with the poor presentation of the work.

There is one problem I see, how are you going to convert the nano sized mixtures of iron and rare earth oxides into the metal, while iron oxide can be reduced with hydrogen gas I do not think that it will be possible to reduce the lanthanides with hydrogen gas. I think you need to make the paper clearer so the reader can see a route from waste magnet back to magnets.

Also why would your process be any better than exposing unwanted magnets to hydrogen gas to convert them to powder before pressing and heating the hydride powder to form a new magnet. I know that in the USA that a method exists for converting the plutonium pit of an atom bomb into a hoop of plutonium metal. The pit is put in the top of the experimental rig where it is exposed to hydrogen, the plutonium hydride particles fall from the pit onto a cone which spreads the solid into a ring. The ring is heated to decompose the hydride back into the metal. This allows the hydrogen to be recycled. What is to stop someone building a version of this gadget for magnets ? I see such a gadget as one which would be far better for making new magnets (as long as you are happy not to change the chemical make up of the magnets. 

Sadly I have to vote for rejection of the paper.

Author Response

Firstly, the authors would like to thank to reviewer and editor for the collaboration in the improvement of our work. The manuscript was revised and corrections were made based on the reviewer’s and editor’s suggestions. The responses to the comments of the reviewers and editor are listed next and the corrections made are highlighted in the revised manuscript.

The great problem with your paper is that your process does not give a separation of the elements in the magnet waste. You process will always give a final product which is the average of the elemental composition of all the magnets which went into the process. I think as a result it is not a good process, if a few magnets which are samarium cobalt were to be accidently added to the neodynium / iron magnets which the process is fed on then the resulting magnet powder from your process would be contaminated with cobalt and samarium. I suspect that the addition of cobalt / samarium to the magnets would have an effect on their magnetic properties.

Without a means of making a separation the recycling industry will make new products which will be lower and lower grade. I know that the Prince of Wales is concerned that nanobots will reduce the earth to "grey goo" but while Charles is concerned about a problem which does not exist. I think that your process will make in the long run the magnet version of "grey goo", each generation of magnets from the recycling process will be worse than the last.

I think that what you need is a separation method which allows you to improve the purity of the Fe and Nd in the solid which is going to be made into new magnets.

Thank you for your invested time and valuabled comment. Firstly I would like to explain some misunderstanding. Our last product consists of rare earth oxide (Nd2O3 and Pr2O3), not iron oxide. First, we separated iron before the production of the last product. To remove iron oxide, firstly nitric acid baking proces was performed with subsequent water leaching. During water leaching, the iron-containing compound was separated as a solid residue. After the removal of iron from leach liquor, RE-oxide (Nd2O3 and Pr2O3),  was produced by ultrasonic spray pyrolysis technique. The iron removal process was explained in detail as follows in the manuscript. In order to better understand our process and separation steps I injected new Figure 1 in our text.

The extraction of REEs from NdFeB magnets was employed by acid baking with HNO3 followed by water leaching. The acid baking process was employed highly concentrated HNO3 (65%) with a 1:5 solid/liquid (S/L) ratio. Some water added to the magnet powders to promote the ionization the nitric acid before acid-baking process. After waiting 1 hour, the mixture was calcined at 200 °C for 2 hours. Water leaching experiments were performed in a 500 mL four-neck glass reactor equipped with a heating mantle and temperature controller. The leaching solution was kept under 550 revolutions/min agitation by a mechanical stirrer. Water leaching experiments were conducted with a 1:15 solid/liquid (S/L) ratio for 90 minutes. The leaching mixture was filtered using the filtering set up to separate the leaching solution from leach residue. Chemical content in the leach liquor was analyzed by ICP-OES to determine the purity of the leaching solution containing REE (see you Table 4; Fe <1 mg/L)

figure 2 has a lot of noise, the quaility of the data and the figure are not very convincing to me. I think that you need to work on the quaility of the XRD data which you present.

We put four new Figures in order to improve Quality of paper. Small quantity of this fine powders was a weakness during XRD-measurement.

your paper is not easy to follow, in figure 5 you show XRD traces for solids made at different tempertures. Rather than writing the tempertures on the graph you use "cryptic codes". Please put an effort in to make the paper more easy to read.

Firstly we offered new Figure 1 with all separated studied steps. Additionally we inserted new picture of experimental Setup, XRD- and  EDS-Analysis.

You the author should put the effort in to making the paper easy to read rather than expecting the reading to make an effort to deal with the poor presentation of the work.

We invested new efforts in to making the paper easy to read and included new Figure 1, Figure 2, Fig. 5, and Fig. 8 to understand our work with different Separation steps. We offered also new reference.

There is one problem I see, how are you going to convert the nano sized mixtures of iron and rare earth oxides into the metal, while iron oxide can be reduced with hydrogen gas I do not think that it will be possible to reduce the lanthanides with hydrogen gas. I think you need to make the paper clearer so the reader can see a route from waste magnet back to magnets.

I have to repeat that our final product is mixture of Rare earth oxide (Nd2O3 and Pr2O3) in the absence of iron. Iron was removed during previous leaching and hydrolysis process. You have right. It is not possible to reduce lanthanides with hydrogen gas. For this purpose we can use molten salt electrolysis. This is traditional technique at the RWTH Aachen University. In our conclusion I inserted  new sentence about molten salt electrolysis of rare earth oxides and one new reference

Cvetkovic, V., Feldhaus, D., Vukicevic, N., Barudzija, T., Friedrich, B., Jovicevic, J. Investigation on the Electrochemical Behaviour and Deposition Mechanism of Neodymium in NdF3–LiF–Nd2O3 Melt on Mo Electrode, Metals 2020,10, 576 DOI:10.3390/met10050576

Also why would your process be any better than exposing unwanted magnets to hydrogen gas to convert them to powder before pressing and heating the hydride powder to form a new magnet.

We need no hydrogen in our work. The solution for metal winning is molten salt electrolysis. The production of new magnets is not our subject. Our main aim is recovery of Nd and Pr from spent magnets and deliver it to Magnet Producer such as Magneti, Ljubljana. Slovenia. They need nanosized powders in order to spare materials and obtain improved magnetic properties.

I know that in the USA that a method exists for converting the plutonium pit of an atom bomb into a hoop of plutonium metal. The pit is put in the top of the experimental rig where it is exposed to hydrogen, the plutonium hydride particles fall from the pit onto a cone which spreads the solid into a ring. The ring is heated to decompose the hydride back into the metal. This allows the hydrogen to be recycled. What is to stop someone building a version of this gadget for magnets ? I see such a gadget as one which would be far better for making new magnets (as long as you are happy not to change the chemical make up of the magnets. 

Treatment with hydrogen can be studied, but this is not subject of this work. We are not magnet producers.  Our next step is a study of molten salt electrolysis of produced mixture of powders (Nd2O3 and Pr2O3), what is final workpackage at Project Call identifier CORNET 29th Call under grant agreement EN03193/20 in 2021/2022 in Germany.

Sadly I have to vote for rejection of the paper.

We respect your final decision. Because of your decision we invested our additional time to explain our new idea to produce fine Nd2O3 and Pr2O3 from spent magnets, what is missing in literature and represents an innovative aspect.